# Korean Nursing Students’ First Experiences of Clinical Practice in Psychiatric Nursing: A Phenomenological Study

**DOI:** 10.3390/healthcare8030215

**Published:** 2020-07-17

**Authors:** Sunkyung Cha, Hyunjung Moon, Eunyoung Park

**Affiliations:** 1Department of Nursing Science, Sunmoon University, Asan 31460, Korea; skc0701@hanmail.net; 2College of Nursing, Incheon Catholic University, Incheon 21987, Korea; hjmoon@iccu.ac.kr; 3College of Nursing, Chungnam National University, Daejeon 35015, Korea

**Keywords:** clinical practice, nursing students, phenomenological study, psychiatric nursing

## Abstract

Nursing students have a more negative attitude toward psychiatric practice than other practices. In particular, Korean nursing students experience increased pressure during clinical practice in psychiatric nursing due to sociocultural and institutional influences, such as prejudices, fear, and anxiety towards mental illnesses. This study aimed to conduct an investigation on students’ first experiences of clinical practice in psychiatric nursing. Participants were 12 fourth year nursing students in South Korea. The data were collected through semi-structured interviews, and data analysis was done using Colaizzi’s phenomenological method. The students’ experiences of clinical practice in psychiatric nursing could be categorized into emotional fluctuation, burnout, transformation, and growth. The results of this study show that nursing students experienced emotional fluctuation and burnout at the beginning of their clinical practice in psychiatric nursing. At the end of the clinical practice, they experienced transformation and growth. The study suggests that nursing instructors and on-site staff need to interact with nursing students to understand the nature of these first experiences and support them through teaching and field guidance.

## 1. Introduction

The length of hospitalization for mental illnesses in Korea is longer than that in other countries [1], and Korea’s involuntary hospitalization rate is the highest among Organization for Economic Cooperation and Development (OECD) countries [2]. Social prejudices regarding mental illnesses are important problems that affect not only the individual, but also the family, community, and country. These prejudices are widely distributed globally, regardless of culture, including Korea [3,4,5]. Negative behaviors from misinformation include the misconception that people with mental illnesses are dangerous, prejudices about interpersonal relationships with those with mental illnesses, and employment discrimination [5].

Nursing students have a more negative attitude toward psychiatric practice than other practices [6]. Nursing students experience increased pressure during clinical practice in psychiatric nursing due to prejudices, fear, and anxiety towards mental illnesses, and show differences in establishing relationships or the satisfaction of clinical practices in psychiatric nursing [7,8,9].

Korean nursing students experience difficulties in establishing efficient relationships with patients because of the gap between theory and praxis. Furthermore, feelings of anxiety and fear and problems developing professionalism due to the absence of role models contribute to these difficulties [10]. Early phenomenological studies on stress experienced by Korean nursing students during clinical experience in a psychiatric unit highlighted the negative side of clinical practices [11]. A recent study showed the importance of the therapeutic environment, the influence of nurses’ capabilities on the quality of the nursing provided, and the relationship between clinical practice and nursing knowledge [9]. Nursing students’ experiences were further affected by environmental factors, such as older facilities and negative and positive situations [12].

In other countries, nursing students in Taiwan break the stigma of mental illnesses through clinical practice, establishing trusting relationships with patients, achieving professional skills and knowledge, and experiencing growth [13]. Nursing students from Saudi Arabia and Iran experience fear and anxiety initially, resulting from a lack of knowledge and experience and from negative attitudes towards mental illnesses [7,8]. In Turkey, communication plays an important role in treating mental illnesses and clinical practice in psychiatric nursing helps to improve interpersonal relationships [14]. Such research suggests that psychiatric clinical practice helps nursing students change their behaviors and attitudes towards mental illnesses and develop a better understanding of themselves and others.

There is a need to investigate Korean nursing students’ experiences of clinical practice in psychiatric nursing. The purpose of this study was to understand the experiences of nursing students during their first clinical practice in the psychiatric unit using a phenomenological approach. This study provides basic data for the development of clinical practice instructions and an education curriculum for psychiatric nursing.

## 2. Materials and Methods

### 2.1. Design

This descriptive and exploratory study used semi-structured interviews to identify the nursing students’ first clinical practice experiences in psychiatric nursing. Data were analyzed using the phenomenological approach method by Colaizzi [15].

### 2.2. Participants

Purposive sampling was used to select participants who could best represent the relevant topic. Participants were fourth year nursing students in Korea who had previously taken a course on psychiatric nursing for two hours a week and had completed 90 hours of clinical practice in psychiatric nursing for 10 days, working nine hours a day for two weeks. A full-time lecturer and a clinical practice instructor led the clinical practice, and there was an on-site instructor as well. Some participants underwent clinical practice in two psychiatric units of two university general hospitals. Some completed their clinical practice in three psychiatric units in one national hospital. Participants were selected until the study reached a saturation point of finding no new descriptions about the study. The final participants of this study were 12 female students with a mean age of 21.

### 2.3. Data Collection

Semi-structured interviews between 60 and 90 min long were conducted with each participant. The interviews had open-ended questions centered on the experiences of the students in clinical practice, allowing the participants to express their subjective thoughts and feelings. The questions of the interviews included: “Please tell me about your experience of clinical practice in psychiatric nursing”, “What did you feel and think during the clinical practice?”, and “What does the clinical practice in psychiatric nursing mean to you?” Prompts were planned and utilized to narrow down each key question. In addition to key questions, additional questions were asked in accordance with the expressions or responses of the participants during the interview. The participants were questioned to elicit in-depth and specific information about their experience and understand their meaning in the context of the interview. Examples of key interview questions and prompts in interviews are shown in Table 1. The interviews were conducted and recorded one to three weeks after their clinical practice ended, and were then transcribed and analyzed. The transcripts were returned to the participants to ensure the quality and accuracy of the data. Secondary interviews with two participants were held regarding statements that needed clarification, consisting of additional questions based on the contents of the first interview.

### 2.4. Ethical Considerations

This study was certified by the Institutional Review Board of the university (No.: SM-201504-006-1). Before the in-depth interviews, we explained the purpose of this study to the participants. After establishing a rapport with the participants, we obtained informed written consent. Two researchers, who were responsible for the clinical practice, rather than evaluating the students, conducted the interview so that the participants could freely express their thoughts and feelings regarding the clinical practice experience. The interviews were recorded with participant consent, and we explained that all contents of the interview were solely for the purpose of this study and would remain anonymous. Students were free to withdraw from participating in the study at any time.

### 2.5. Analysis

Data analysis was conducted using the phenomenological method [15]. First, we explored the meanings by transcribing the direct words of the participants and slowly reading them repeatedly and independently. Next, we deduced significant statements related to the experiences of the nursing students from each transcript. Third, we identified the meanings from these significant statements. The fourth step was to structure the comprehensive meanings into themes. We discussed the themes and planned how to deduce them and their names until reaching a consensus. The fifth step was to compare the transcripts and the themes multiple times to secure the justification of the results. Sixth, we worked to support the themes and the thematic techniques with the statements to secure the justification. Finally, we confirmed whether the themes and the contents, including quotations, matched what the participants had intended by consulting with two interviewed nursing students.

To ensure the trustworthiness and credibility of our results, two of the participants confirmed the meanings, theme clusters, and themes derived from the data analysis. For dependability, all the interviews were recorded and transcribed so that the original data could be referred to. To ensure transferability, the results and the context were described extensively. For confirmability and to eliminate bias as much as possible, the two researchers independently analyzed the data and confirmed the meanings. They then discussed any disputed themes until there was a consensus. This was also established by meeting the three standards discussed previously.

## 3. Results

The results were reduced to four theme clusters and twelve themes (Table 2).

### 3.1. Emotional Fluctuation

The participants were anxiously watchful towards the patients and were confused as to how to react to the patients’ behaviors. During this process, the students felt a lack of sufficient skills and experienced an indescribable emotional fluctuation and anxiety that could not be explained in one sentence. These feelings were more common at the beginning of the clinical practice.

#### 3.1.1. Watchfulness

The participants showed great watchfulness towards the patients at the beginning of the clinical practice. They did not have any information on the patients and rapport was not established before the clinical practice. Participants were anxious and stressed due to uncertainty. At the same time, some students were directly or indirectly exposed to violent and uncontrollable patients, which led to great fear. They hoped that clinical practice would be over soon so that they could avoid the discomfort and unfamiliarity:


*Anxiety. You don’t know what may happen behind your back. There was that kind of anxiety. I was uncomfortable when I talked to the patients, and was constantly careful, as I did not know how the patients would react… It was difficult due to the anxiety; I was stressed and only waited to go home.*
*(Participant 3-1)*


*I was really scared… (The patient) was playing ping-pong with me, and then suddenly grabbed another person by the collar and threw the person to the ground, and I was really scared and thought that my heart would drop. I thought that I should be careful when near the patient.*
*(Participant 7-1)*

#### 3.1.2. Confusion

The participants explained that they were confused and found it difficult to adapt to the clinical practice. They also had mixed emotions of amusement, fun, and enjoyment, which was different from what they experienced in other practices. For example, they did not know how to react to some of the patients’ difficult requests. Similar confusion emerged when they were embarrassed or angry at patients who made sexual jokes or tried to make physical contact. The students even felt a sense of guilt, which led to emotional fluctuation during the clinical practice, followed by confusion.


*In the first week, I was kind of confused and wondered what I should do, I had to adapt but could not adapt and so on. The first week was just simply difficult.*
*(Participant 4-1)*


*There were various emotions during the clinical practice. It was weird and amazing at the same time, and sometimes it was difficult to decide how to react. However, sometimes it was fun and enjoyable…I thought it would be mentally difficult, but there are various aspects integrated.*
*(Participant 10-1)*

#### 3.1.3. Immaturity: Lack of Quality Skills

The participants reflected on their immature knowledge, emotions, and reactions and expressed their lack of quality skills as a semi-therapist. Many students were not confident that their actions would aid the patients they wished to help. Some were dishonest to the patients, leading to conflicts when crossing the therapeutic boundaries and limitations of countertransference.


*I wanted to try my best for the patients, and I wanted to do what the patients wanted the most, but it was hard when it was difficult to decide whether this was the best solution… I was so immature, so that was hard. I thought it was best to do what the patients felt happiest, but what I learned at school clashed with what I thought at the time, so I did not know what was right.*
*(Participant 11-2)*

### 3.2. Burnout

In their first encounter with a patient, students felt extreme stress and, subsequently, great emotional exhaustion. They were also physically drained during their meetings. The students even felt helpless as they thought they were alone and could not receive help from anyone.

#### 3.2.1. Emotional Outburst

The participants went through emotional outbursts from the accumulated stress in establishing rapport with patients. Burdensome assignments during the clinical practice and the efforts to unilaterally connect with the patients also contributed. They recognized the outbursts only after the completion of clinical practice, when they thought about the things that had transpired during said practice.


*I felt a great emotional outburst. It did not feel good returning [home after the end of the clinical practice]… I was emotionally exhausted, and the things that happened there did not stop there. It was hard since they followed me home and made me think about it even when I was home.*
*(Participant 4-1)*


*During the last conference (after the clinical practice,) I suddenly burst out in tears when talking about what happened with the patients… I did not think it was much of a problem. I thought that after it is over, it would be nothing so I smiled, but when I was tearing up in the end, I felt ‘Oh, it was all accumulated inside’.*
*(Participant 1-1)*

#### 3.2.2. Exhaustion

While the participants tried to maintain a therapeutic relationship with the patients, accumulated stress eventually led to emotional and physical exhaustion. This created a negative impression for some students of the clinical practice.


*Overall, I only have bad memories, and I always wished that the clinical practice would be over soon…Of course, it was not physically difficult, but it was not mentally easy…[It was strenuous] day by day because of the patient who made me suffer.*
*(Participant 1-1)*

#### 3.2.3. Helplessness

The feeling that they were alone in their stress and had no one to open up to and share the burden with made the participants feel helpless. They could not talk about it with their friends or the nurses because the thought that they could not be trusted led them to think they were unprotected. Furthermore, the students felt that they were emotionally locked up due to the secure and protected nature of the psychiatric in-patient ward.


*When I feel stressed, I have to receive empathy and get rid of it… I could not get rid of the stress. I did not want to talk about it with my friends, and there is stress that comes from this place. I cannot talk about with my best friends. […] The head nurse did not trust me…The medical staff did not help me. I think I got teary as no one would rescue me, and I could not be protected.*
*(Participant 3-1)*


*I felt like I was locked up in the ward. It is a closed ward, so from the minute I went in…I felt a lot of emotional burnout.*
*(Participant 4-1)*

### 3.3. Transformation

Participants also naturally broke their negative prejudices towards psychiatric patients by communicating with them. Participants truly understood the patients through thoughtful emotional exchanges. At the end of the clinical practice, many participants established emotional attachment with the patients and were extremely disappointed that the clinical practice had to end. Their outlook and emotions towards the patients completely changed.

#### 3.3.1. Breaking Barriers

During the clinical practice, the participants eliminated their prejudices towards patients in the psychiatric unit, and understood that the patients who seemed different were actually ordinary people like them. They came to understand that mental illnesses can affect anyone. The participants explained that the clinical practice broke barriers when it came to understanding mental illnesses and their patients:


*I had strong prejudices against those with mental illnesses, but they broke down a lot. I think that the people with mental illnesses are not much different from us, and the experience led me to have a wider viewpoint.*
*(Participant 12-1)*


*I thought that it [mental illnesses] is as common as the cold, as anyone could be diagnosed…As such, the symptoms are likely to be alleviated a lot as well.*
*(Participant 2-1)*

#### 3.3.2. Empathy

The participants had the opportunity to talk to the patients and spend more time with them. Rather than having perfunctory relationships, the efforts of the nursing students to truly understand the patients led to a deep empathy for the patients’ painful past experiences. They felt that these emotional exchanges and communication had become the foundation of true psychiatric nursing care.


*Rather than simple perfunctory meetings… I was happy when they sympathized with me and talked with me… (If I) understand them first then maybe it will be possible to create real nursing care. (…) Really understanding the others and really being worried about them.*
*(Participant 3-1)*

#### 3.3.3. Jeong: Emotional Attachment

At the end of the clinical practice, the participants reported that they had a special affection and longing towards the patients, unlike in the beginning. Participants created emotional attachment, called a sense of “jeong” in Korea, as they did not want to part with the patients. They expressed that they often think of the patients and occasionally miss them. Unlike the other clinical practices, they felt a feeling of worry, as if they left behind precious children.


*I did not want to leave after the end of the clinical practice. I became so attached to the patients and I still think about them, I miss what I did with the patients until now and I want to see them again. (…) I had more Jeong (emotional attachment) with them than any other patients from other clinical practices... because I was with them day and night.*
*(Participant 11-1)*

### 3.4. Growth

The participants recognized that the clinical practice allowed them to reflect, understand themselves better, and be thankful for their lives. Not only did the participants provide help to the patients, but they also felt that they changed and healed themselves through the patients.

#### 3.4.1. Introspection

The clinical practice became an opportunity for students to reflect on themselves and review their lives through interactions with patients. In particular, the clinical practice became an opportunity for students to see their weaknesses and reflect on a more positive outlook towards life:


*I looked back on myself a lot…I was not satisfied with my life and had a lot of complaints, but I had introspections through the patients who were thankful for the small things although they did not have much. That was a big thing for me.*
*(Participant 12-1)*

#### 3.4.2. Appreciation

The participants learned to appreciate their lives after going through attitudinal changes after the clinical practice. They expressed that they realized how fortunate they were compared to the patients they encountered. In addition, the clinical practice made them realize that the small things that they did not consider before were blessings and sources of happiness:


*[There was a] change…in my attitude. I think I got frustrated really easily when something did not go as I wanted it to. However, during my clinical practice, I came across many others who were going through much more difficult times than I am. That caused me to think ‘I am happy’, and made me appreciate and be satisfied with what I have compared to those who are ill, which led to the changes.*
*(Participant 11-1)*

#### 3.4.3. Maturity

The participants felt that the clinical practice provided an opportunity for them to heal both themselves and the patients. It also allowed them to become more mature and more stable. These experiences affected not only their relationship with the patients but also their relationships with others. This motivated them to pursue careers as psychiatric nurses.


*I think the experiences contributed to my growth… and now that I look back from afar, it was a time when I shined. I became more mature and believe that I was some help to them and it made me think how we should live to become happier…[It] made me think about what I should do, and that I wanted to become a psychiatric nurse.*
*(Participant 3-1)*

## 4. Discussion

### 4.1. Emotional Fluctuation

The emotional fluctuation at the beginning of the clinical practice in psychiatric nursing is in line with the fact that nursing students experience anxiety and sometimes emotional wounds due to prejudices, regardless of their theoretical knowledge or other clinical practice experiences [7]. In the beginning, they felt personal limitations and decreased capabilities during their first clinical practice [10], as well as fear and anxiety [7,8]. The results of this study indicate that, despite the periodical changes, nursing students’ early experiences with mental illnesses or mental hospitals did not seem to change. Given past studies that have found theoretical education insufficient to eliminate the stigma of mental illnesses before or after clinical practice [14], the education curriculum should be tailored to provide sufficient information on the psychiatric patients and the therapeutic environment before the clinical practice. The emotional burdens of nursing students can be alleviated through proper education on the therapeutic relationship and communication in practical psychiatric nursing situations. Curricula should also explain that emotional fluctuations at the beginning of the clinical experience is a normal phenomenon. Detailed methods for reacting to the fluctuation before and during clinical practice should be discussed. The on-site instructor and the lecturer on clinical practice should be aware of these students’ experiences at the beginning of the clinical practice. They should counsel and support them in advance, assist them in preventing these fluctuations, and help them overcome these experiences through positive experiences with practice.

### 4.2. Burnout

Burnout is the physical, emotional, and mental state of weariness when exposed to a situation with significant emotional demands [16]. Psychiatric nurses experience greater job anxiety and burnout compared to general ward nurses [17]. Since nurses feel helplessness and burnout when they do not derive value and meaning from their job, nurses’ internal factors are as important as the external factors when contemplating ways to relieve burnout [16]. Although burnout experienced by nursing students may be different from the burnout of psychiatric nurses, it may be helpful to strengthen the internal factors of nursing students to relieve burnout of any kind. The curriculum needs to clearly define the roles of the nursing students within clinical practice in psychiatric nursing through regular conferences or education for individuals or small groups. Activities to reconfirm their personal value and meaning as semi-therapists will also help reduce feelings of burnout.

In terms of alleviating helplessness, the on-site instructor and clinical practice lecturer are critical resources. On-site instructors should convince the students that they are there to listen and that they trust and support them. They should also help students find the appropriate direction to take. Clinical practice lecturers should support students and talk about the difficulties that the students experience during clinical practice. The burden of psychiatric nursing assignments has long been a problem [18], and lecturers should limit the number of assignments required. It should also be recognized that the on-site instructor and the full-time lecturer of the clinical practice are not evaluators, but educators that assist and support nursing students.

### 4.3. Transformation

Transformation is defined by a transformation of maturity, which nursing students experience through clinical practice in psychiatric nursing regardless of whether there is a good location, good people, and good psychiatric nursing. This form of change differs from the positive attitudinal changes towards the psychiatric unit encouraged by high-tech hospitals, professional psychiatric nursing, and meeting good patients during the nursing students’ first clinical practices in psychiatric nursing [12]. Therefore, transformation is akin to maturation, reflected by students acquiring new techniques, adopting humanitarian behavior, and changing attitudes towards psychiatric patients [8].

Although existing research has identified negative themes of the clinical practice experience in the psychiatric unit, such as a lack of integration of theory and praxis, the absence of a holistic approach to psychiatric nursing, inadequate professional support, and a lack of resources [19], this study uniquely identified a transition from negative experiences to positive outcomes, a finding that is absent from existing research.

Breaking prejudices through understanding mental illnesses and their patients was a major theme in other studies [13,14]. To alleviate the prejudices toward mental illnesses, there should be education and re-education to establish an appropriate awareness and attitude towards those with mental illnesses [20]. Active education and opportunities to frequently come into contact with those with mental illnesses [21], a systematic education through videos and standardized patients [22,23], and training on human rights sensitivity [20] all assist in breaking the prejudices toward those with mental illnesses.

Empathy is the ability to feel another’s emotions and experiences from their perspective. This can be divided into cognitive empathy, which involves accepting the position of the other person and predicting their actions, and emotional empathy, which experiences emotions on behalf of the other person [24]. Communication and empathy through emotional exchanges is a technique of therapeutic communication, and the students who participated in this study reported a genuine, empathetic understanding of the plight of the patients through the clinical practice in psychiatric nursing.

In addition to empathy, the students reported experiencing jeong, an important Korean value, different from love, friendship, or countertransference found in Western cultures. Jeong is experienced through interpersonal relationships and is the common sentiment that defines interpersonal relationships [25]. The participants experienced jeong, as evidenced by their unwillingness to part with the patients, their affection for the patients, and missing the patients after the clinical practices ended. As interpersonal concepts defined according to the values of Western cultures do not apply directly to Korean culture, curriculum designers and implementers should consider the cultural characteristics of Koreans that are implicated when nursing students establish therapeutic relationships during clinical practices. Here, jeong can be used as an advantage or resource. However, if the influence of jeong becomes too large, problems may emerge if a clear division between the patient and the therapist is not maintained, thereby accelerating burnout.

### 4.4. Growth

Growth refers to student self-development while providing nursing care and is a unique outcome of clinical practice in psychiatric nursing. The theme of growth was also found in other studies on the forms of maturation and personal development [8,13,14]. Ultimately, students experienced growth during their first clinical practice in psychiatric nursing, and provided them with an opportunity to further improve as a nursing care provider.

Experiencing appreciation through a relative comparison of their lives with the patients’ lives contributes to a positive experience in psychiatric clinical practice. The theoretical education of psychiatric nursing emphasizes introspection by the nurses before even meeting patients and applying their psychiatric nursing skills. The fact that the students reflected on their experiences and became more aware of themselves and others through clinical practice is a positive outcome, one echoed by previous studies [13]. Introspection and appreciation can serve as positive factors motivating future nurses to pursue their best selves, both at work and in their everyday lives.

### 4.5. Limitations

This study only analyzed the experiences of nursing students working in psychiatric inpatient units in Korea. Therefore, the results of this study may not apply to cultures outside of Korea. Future research should consider the clinical practice experiences of nursing students in other contexts, identifying the specific sociocultural and institutional factors that shape these experiences. Clinical practice in psychiatric nursing in Korea is worth two to four credits for those in their third or fourth year, and consists of programs two to four weeks long [12]. The students participate in clinical practices in: psychiatric units of general, university, national, and public hospitals; private mental hospitals; community mental health centers. Usually, nursing students learn about inpatients. Participants of this study also experienced psychiatric inpatient-related practice; therefore, it may differ from experiences in other settings. In the future, it will be necessary to extend to participants who have practiced in an outpatient or community-based setting.

## 5. Conclusions

This phenomenological study examined the first experiences of nursing students in clinical practice in psychiatric nursing. The nursing students studied here experienced emotional fluctuation as feelings of watchfulness, confusion, and a lack of sufficient skills in the beginning of their clinical practice due to the fear and anxiety of the psychiatric nursing setting. The students also experienced burnout during the practice, as shown by their emotional outbursts, exhaustion, and helplessness. However, by the end of the clinical practice, they broke their prejudices towards mental illnesses and experienced transformation, reflected in their emotional attachment, or jeong, with the patients. Thereafter, the students experienced growth through introspection, appreciation, internal maturity, and positive emotions. Clinical practice lecturers, on-site leaders, and psychiatric nurses need to interact with nursing students to understand the nature of these first experiences and support them through teaching and field guidance.

This study may contribute to an in-depth understanding of nursing students’ first clinical experiences. Nursing educators and nurse practitioners need to understand that students undergo phases of emotional fluctuation, burnout, transformation, and growth. This study can be used to educate and support students and contribute to alleviating the burdens of the students and increasing the quality of clinical practice education.

## Figures and Tables

**Table 1 healthcare-08-00215-t001:** Examples of key interview questions and prompts in interviews.

1. Please tell me about your experience of clinical practice in psychiatric nursing. Prompt: How did you feel when you first met psychiatric patients? How was your interaction with them during clinical practice? What do you think is a useful experience of clinical practice? Had there been any problems or challenges for you? 2. What did you feel and think during the clinical practice? Prompt: What kind of experience did you think you would have before the clinical practice, and how did you feel after the clinical practice? 3. What does the clinical practice in psychiatric nursing mean to you? Prompt: What was the change you felt in yourself before and after the clinical practice? What has been your overall impression of clinical practice?

**Table 2 healthcare-08-00215-t002:** Korean nursing students’ first experiences of clinical practice in psychiatric nursing.

Theme Clusters	Themes
Emotional fluctuation	Watchfulness
	Confusion
	Immaturity: Lack of quality skills
Burnout	Emotional outburst
	Exhaustion
	Helplessness
Transformation	Breaking barriers
	Empathy
	Jeong: Emotional attachment
Growth	Introspection
	Appreciation
	Maturity

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
