# Peer review of "Korean Nursing Students’ First Experiences of Clinical Practice in Psychiatric Nursing: A Phenomenological Study"

_healthcare, 2020, doi:10.3390/healthcare8030215_

Round 1

Reviewer 1 Report

This is a fairly well written paper reporting a qualitative study of nursing students' experiences during their first psychiatric/mental health care clinical rotation in Korea. Below find my detailed comments/suggestions:

  1. In the Materials and Methods section's Participants' subsection, clarify which sampling procedure (presumably convenience sampling) was used and why, for for the first interviews and for the second (checking themes with 2 participants) interviews. 
  2. In the Materials and Methods section's Data Collection subsection, clarify what if any prompts were planned and used for each qualitative question in the semi-structured interview. 
  3. In the Materials and Methods section's Data Collection subsection, clarify whether interview transcripts were validated by a person who did not transcribe them (for standard data quality assurance), and if not why.
  4. In the Discussion section, add at its end a Limitations subsection or paragraph, e.g., addressing the likely under-representation of Korean nursing students and mental health care learning (inpatient, daycare, outpatient and outreach) settings in this study. 
  5. In the Conclusions section, qualify or revised the recommendations for educational practice and policy, as basing such recommendations on such an exploratory study is too preliminary and more research (with more nursing students and learning settings in Korea) is required before such recommendations are addressed. 
  6. The English in the paper has to be moderately improved, e.g., re grammar and syntax in some sentences.    

Reviewer 2 Report

This is an interesting and important manuscript. I have just a few suggestions:

  1. in the introduction, please spell out OECD - I know what it means but everyone might not.
  2. In the second to last sentence under the conclusion (1st sentence in last paragraph), please break up this very long sentence to improve its clarity. I believe I know what you are attempting to convey but it's not clear, further, it would benefit from a little expansion because it's very important. 
